# Study on the Corrosion Behavior of D36 Steel Plate and H62 Copper Alloy Net for Marine Aquaculture Facilities in Simulated Seawater

**Fengfeng Gao [1], Fukun Gui [2], Dejun Feng [2] 🄳, Xiaoyu Qu [3], Fuxiang Hu [4] and Xu Yang [2,*]**

[1] Marine Science and Technology College, Zhejiang Ocean University, Zhoushan 316022, China; gff3489385383@163.com
[2] National Engineering Research Center for Marine Aquaculture, Zhejiang Ocean University, Zhoushan 316022, China
[3] School of Fisheries, Zhejiang Ocean University, Zhoushan 316022, China
[4] Department of Marine Biosciences, Tokyo University of Marine Science and Technology, Tokyo 108-8477, Japan
[*] Correspondence: 2022188@zjou.edu.cn

**Abstract:** Marine aquaculture facilities have been working in a high salt and humidity marine environment for a long time, which makes them be inevitably affected by seawater corrosion, especially the main structures such as metal mesh and frame. Therefore, studying the corrosion behavior of net and frame steel is of great significance for the selection of materials and corrosion protection of marine aquaculture facilities. The influence of NaCl concentrations and immersion state on self-corrosion behavior and the influence of layer thickness and overlapping area on galvanic corrosion behavior of H62 copper alloy mesh/D36 steel plate was discussed in this study using weight loss and electrochemical measurements. The tensile tests were conducted to observe the influence of different corrosion conditions on maximum force and tensile strength of the net. The corrosion rate of the two materials increased rapidly with the increase of NaCl concentrations; the corrosion rate of both materials showed the decreasing trend with the extension of time, and the corrosion rate of H62 was always lower than D36 steel. When two materials were coupled, the galvanic corrosion rate would decrease with the increase of the layer thickness and overlapping area. The tensile results were consistent with the corrosion results. When these two materials work together, adding layer thickness or increasing the overlapping area is one of the ways to protect the frame steel to a certain extent.

**Keywords:** marine aquaculture facility; copper alloy net; simulated seawater; galvanic corrosion

## 1. Introduction

In order to cope with the increase of the population's demand for protein and food security brought by the population growth in the world and economic development, aquaculture has developed rapidly. The total global aquaculture production rose to the highest level of history in 2020, and the world aquaculture production increased from about 10 million tons of live-weight seafood in 1990 to 214 million tons in 2020 [1]. There are many mariculture modes, such as raft cultivation, net cage aquaculture, pile-net enclosure culture and so on [2–4]. Among them, the net is a very important component, which is the guarantee of safety and production effect of mariculture [5]. Furthermore, the expansion and intensification of offshore mariculture have also exerted great pressure on the marine environment, and offshore fishery resources continued to decline, which restricted the development of the marine farming industry [6–9]. Facing the complex deep-sea environment, the traditional net cannot meet demand because of poor safety performance, poor anti-fouling effect, net damage, and poor water exchange capacity [10,11]. Copper alloy net not only has strong wind and wave resistance and antibacterial and anti-biological adhesion ability, which can provide a safer and healthier culture environment, but also

is easy to recycle, which makes copper alloy net replace the traditional synthetic fiber net as the first choice for "healthy, safe, and environmental protection" development of aquaculture [12,13]. Therefore, the copper alloy net was studied by more researchers.

In recent years, scholars have conducted a lot of research on copper alloy net. Nie et al. [14] studied the performance of rigid–flexible copper alloy expanded net by hydrodynamic model tests and found that the resistance of copper alloy expanded net in water was higher than that of polyethylene fiber net, and the economic efficiency was also higher than the latter. Wang et al. [15] tested the resistance of different copper wire diameters and mesh sizes in a water tank at different trailer speeds and found that the resistance increased with the increase of wire diameter and mesh size. Wang et al. [16] conducted experiments on the fatigue performance of copper alloy net structures to provide a basis for the assessment of copper alloy net structures' safety. Tyler et al. [17] observed chemical parameters in water and sediment from brass net cages installed in Panama and Mexico, revealing that brass nets in the ocean had no impact on either the marine environment or cultured organisms. As mentioned above, the current researches on copper alloy net are mostly focused on anti-fouling, hydrodynamics, etc.

Lots of copper alloy nets have been used in the marine aquaculture facilities (shown in Figure 1). Marine aquaculture facilities work in high salinity, high humidity, and other complex marine environments for a long time under the full-immersed or semi-immersed state, so the net and frame are highly susceptible to be corroded by seawater, which seriously affects the functionality and safety of the net structure [18]. Therefore, the studies of the corrosion behavior of net and frame steel in seawater are beneficial to fully learn the material corrosion conditions, accumulate material corrosion data, and propose measures for material corrosion protection timely and accurately to avoid materials failure due to corrosion.

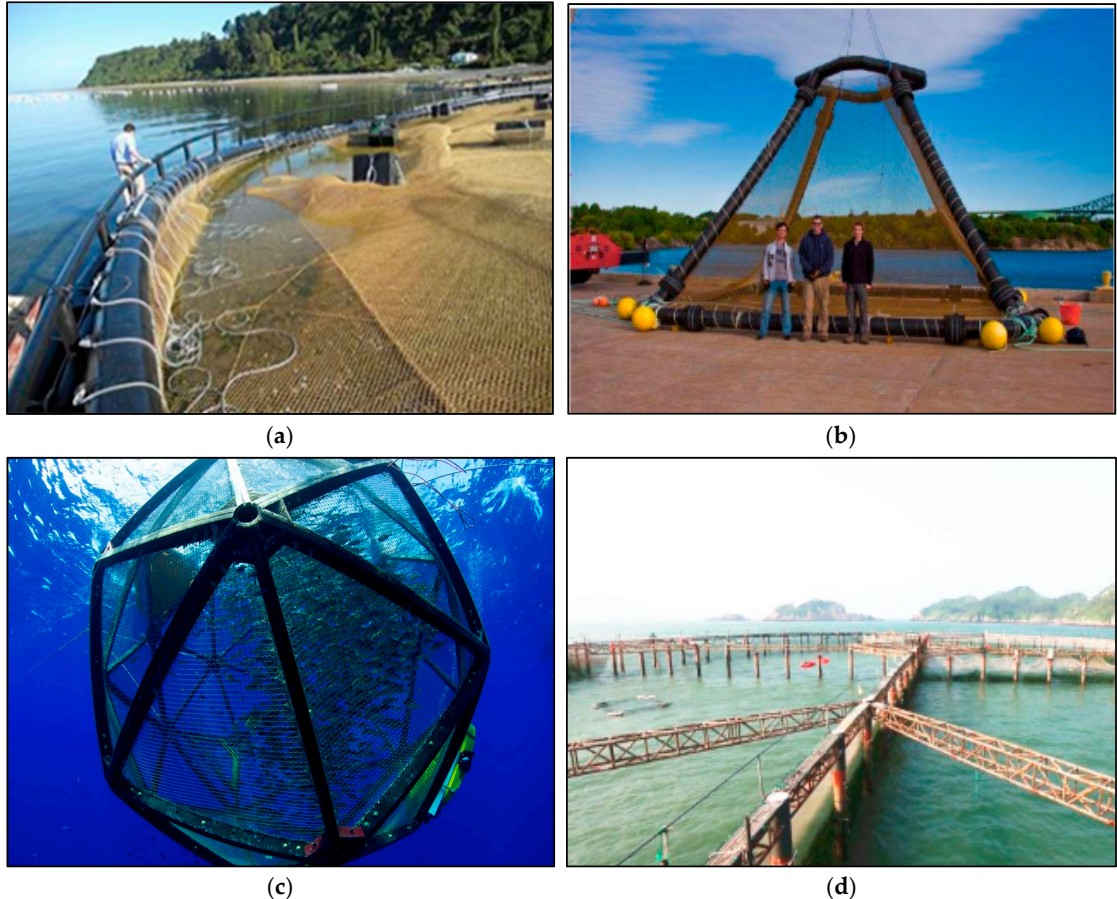

(a)　　　　(b)

(c)　　　　(d)

**Figure 1.** Copper alloy application of marine aquacultures. (**a**) China copper alloy net cage. (**b**) OCAT copper alloy net cage. (**c**) Aquapod copper alloy net cage. (**d**) Copper alloy pile-net enclosure.

It can be easier to control the marine environment variables, get the mechanism of material corrosion, obtain and analyze the experimental data faster via laboratory experiments. Currently, there are many laboratory studies on H62 copper and D36 steel. Zhang et al. [19] conducted weight loss tests and electrochemical tests on H62 and HAl67-2.5 in static artificial seawater, finding that the corrosion rate of aluminum brass was significantly lower than that of H62, and both had a significant tendency of galvanic corrosion. Zhang et al. [20] studied the galvanic corrosion behavior of alloy 5083 and H62 under different magnetic forces. It was found that the magnetic force had a great influence on galvanic corrosion. The higher the magnetic force, the greater the corrosion rate. The presence of magnetic fluid led to the effect of the magnetic force on corrosion. Wang et al. [21] studied the atmospheric corrosion of two types of coppers exposed to the urban environment, finding that the corrosion rate of T2 increased first and then decreased; the corrosion rate of H62 kept decreasing, and the corrosion rate of H62 was better than that of T2. Lu et al. [22] studied the corrosion behaviors of T2 and H62 in the simulated Nansha marine atmosphere. The results showed that the marine atmosphere had a very strong influence on the corrosion of both types of copper; the dezincification corrosion of H62 was very obvious, and the corrosion products were mainly zinc-rich compounds. Yang et al. [23] studied the galvanic corrosion behavior of magnesium alloy and copper in distilled water and found that the corrosion rate of magnesium alloy was enhanced when these two metals were coupled, and the temperature and immersion time accelerated corrosion. Tang et al. [24] designed a new CuNiMnCrAl alloy and tested its corrosion behavior in a 3.5% NaCl solution, finding that the new alloy had a nice corrosion resistance because of the $Al_2O_3$ oxide film on the surface. In this paper, the corrosion behavior of the H62 mesh/D36 steel plate under full/semi-immersed state was studied in NaCl solutions mainly by means of the weight loss method, polarization curve, electrochemical impedance spectroscopy (EIS), etc. The influence of different bedding layer thicknesses, and overlapping areas on the galvanic corrosion behavior of the mesh/frame steel under the semi-immersed state was also studied in order to provide data support for the selection of net materials, reduce corrosion rate effectively, and improve the service life of materials.

## 2. Materials and Methods

### 2.1. Experimental Materials

The experimental materials were the D36 steel plate and H62 copper alloy mesh, which was made with copper alloy wire by a weaving machine. The structures of wire and mesh are shown in Figure 2. The main chemical compositions of materials are shown in Table 1. The conditions of the experiments are in Table 2.

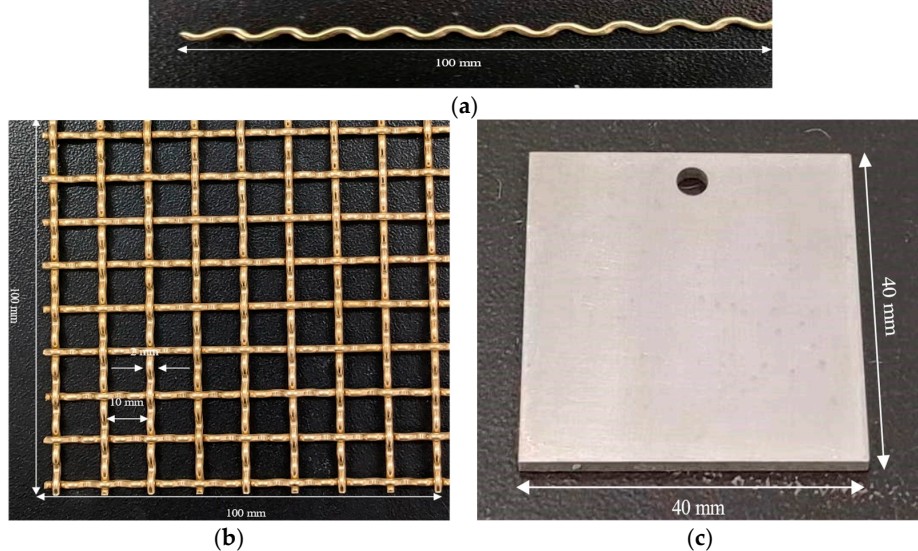

**Figure 2.** Experimental materials. (**a**) Copper alloy wire structure. (**b**) H62 copper alloy mesh. (**c**) D36 steel plate.

**Table 1.** Main chemical compositions of the two materials (mass fraction, %).

| Materials | Cu | Mn | P | S | Si | Cr | C | Ni | Fe | Pb | Zn |
|-----------|------|------|-------|-------|------|-------|------|-------|------|-------|------|
| H62 | 61.82 | - | - | - | - | - | - | 0.28 | 0.08 | 0.015 | Bal. |
| D36 | 0.018 | 1.38 | 0.016 | 0.006 | 0.25 | 0.021 | 0.15 | 0.008 | Bal. | - | - |

**Table 2.** Conditions of experiments.

| Case | NaCl Concentration | Immersion State | Layer Thickness (mm) | Overlapping Area (cm$^2$) |
|------|--------------------|-----------------|----------------------|----------------------------|
| 1 | 1.5 | semi-immersed state<br>full-immersed state | - | - |
| 2 | 2.5 | semi-immersed state<br>full-immersed state | - | - |
| 3 | 3.5 | semi-immersed state<br>full-immersed state | - | - |
| 4 | | | 5 | - |
| 5 | | | 10 | - |
| 6 | 3.5 | semi-immersed state | 15 | - |
| 7 | | | - | 28 |
| 8 | | | - | 52 |
| 9 | | | - | 100 |

*2.2. Sample Preparation and Pretreatment*

The influence of NaCl concentration, immersion state on the self-corrosion behavior, and the influence of layer thickness and overlapping area on galvanic corrosion behavior of H62 copper alloy mesh/D36 steel were studied in this paper by weight loss and electrochemical measurements. Tensile tests were conducted on the corroded mesh wire to observe the influence of corrosion on the maximum force and tensile strength of copper wires. The preparation, pretreatment of samples, and research methods were as follows:

2.2.1. Single Metal Immersion Experiment

The specimen size of H62 copper alloy mesh was 100 mm × 100 mm; the mesh size was 10 mm, and the wire diameter was 2 mm. The surface treatment process was as follows: soak the meshes in the acetone for 10 min to remove oil; rinse with ethanol absolute, ultrapure water, and deionized water; blow dry them with a hair dryer; pickle with 15% sulfuric acid solution at room temperature for 3–5 s; place the meshes into deionized water; ultrapure water for ultrasonic cleaning twice; use deionized water; ultrapure water to rinse the meshes again; blow dry meshes with cold air and place them in a drying oven for 24 h to be used. D36 steel plate size was 40 mm × 40 mm × 2 mm. The surface treatment process was as follows: burnish with 400#, 600#, 800#, and 1200# water sandpaper step by step; polish with polishing paste, then remove the oil with acetone; rinse with ethanol absolute; clean the plates for ultrasonic cleaning with ultrapure water and deionized water; place the specimen into a desiccator after blow drying with cold air. Parallel specimens in each condition of at least three specimens.

2.2.2. Galvanic Corrosion Experiment

The size of the mesh was 135 mm × 135 mm × 2 mm; the steel plate size was 100 mm × 100 mm × 2 mm; the treatment was the same as the single metal immersion experiment; the redundant area was encapsulated with epoxy resin.

2.2.3. Electrochemical Test

The size of H62 copper alloy mesh was 10 mm × 10 mm × 2 mm; steel plate sizes were 10 mm × 10 mm × 2 mm, 40 mm × 40 mm × 2 mm, and 100 mm × 40 mm × 2 mm. Firstly, the specimens were sealed in the mold with epoxy resin after welding copper wire on one end of the specimen and energizing it with a multimeter. Secondly, the working surface

(the other side opposite to the welding) was polished with 400#, 600#, 800#, and 1200# water sandpaper step by step, polished with polishing paste, removed oil with acetone and ethanol absolute, then cleaned with ultrapure water and deionized water after the epoxy resin solidified. Finally, dry them with cold air and put them into the dry dish for later use.

### 2.3. Experimental Equipment and Measurement

### 2.3.1. Single Metal Immersion Experiment

The mass of specimens before corrosion was weighed with an analytical balance of type FA2004N (accuracy 0.1 mg). Immersion states of materials are in Figure 3. PVC pipe was used for support to hang the specimens with nylon rope in plastic containers which contained 1.5%, 2.5%, and 3.5% NaCl solutions. The pH was 8.2, adjusted by HCl and NaOH. To avoid affecting the experimental results, the experimental temperature was $25 \pm 1\ ^{\circ}C$, and the experimental periods were 120 h, 240 h, and 480 h, with a change of solution every day. All specimens were greater than 3 cm from the bottom of the container and 5 cm from the water surface. Specimens were removed after the experiment, dried, and weighed according to GB/T10124-1988. Corrosion weight loss measurement was used to study the mass change of the specimens in different NaCl solutions.

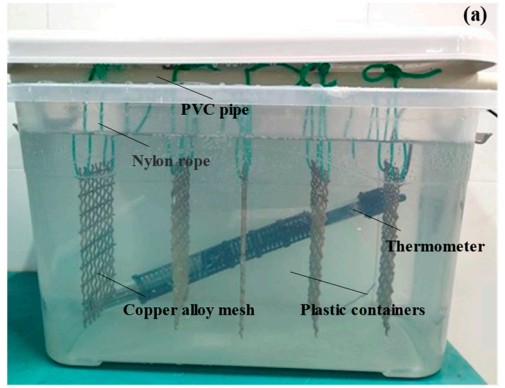 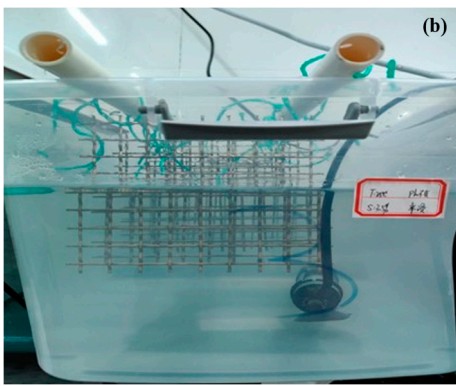

**Figure 3.** Immersion state of H62 copper alloy mesh; (**a**) Full-immersed state; (**b**) Semi-immersed state.

The average corrosion rate of metal can be obtained by Formula (1):

$$V = \frac{8.76 \times 10^4}{7.8 \times s \times t} \times \Delta W \tag{1}$$

where, $V$ is the corrosion rate, mm/a; $\Delta W$ is the average mass loss of the specimen corrosion, g; $S$ is the immersion area of the specimen, $cm^2$; $t$ is the immersion duration, h.

### 2.3.2. Galvanic Corrosion Experiments

When two materials with different potentials are used together in the ocean, there is more severe galvanic corrosion than they are used alone. Metals with high potential are protected as anodes, and the corrosion rate decreases, while the corrosion rate of metals with low potential accelerates.

FA2004N Analytical balance (with an accuracy of 0.1 mg) was used to weigh and record the mass of samples before corrosion. The diagram of the H62/D36 galvanic couple with different bedding layers is shown in Figure 4. Different thicknesses of bedding layers (5 mm, 10 mm, 15 mm) were added between the copper mesh and the steel plate. A plastic board was used to fix the lower connection of the steel plate, plastic bedding layer, and copper alloy mesh together. The upper part was fixed with ribbon; finally, nylon rope was inserted above, and galvanic couples were placed in the 3.5% sodium chloride solution in a full-immersed or semi-immersed state. Sodium chloride solution was prepared by deionized water and sodium chloride; the temperature of the solution was 25 °C, and pH was 8.2, adjusted by HCl and NaOH. The semi-immersed test was mainly carried

out by marking the samples, with three parallel samples for each working condition. In experiments with different overlapping areas (Figure 5), the plastic bedding layer was removed, and the couple was overlapped according to the projected areas of 28 cm$^2$, 52 cm$^2$, and 100 cm$^2$.

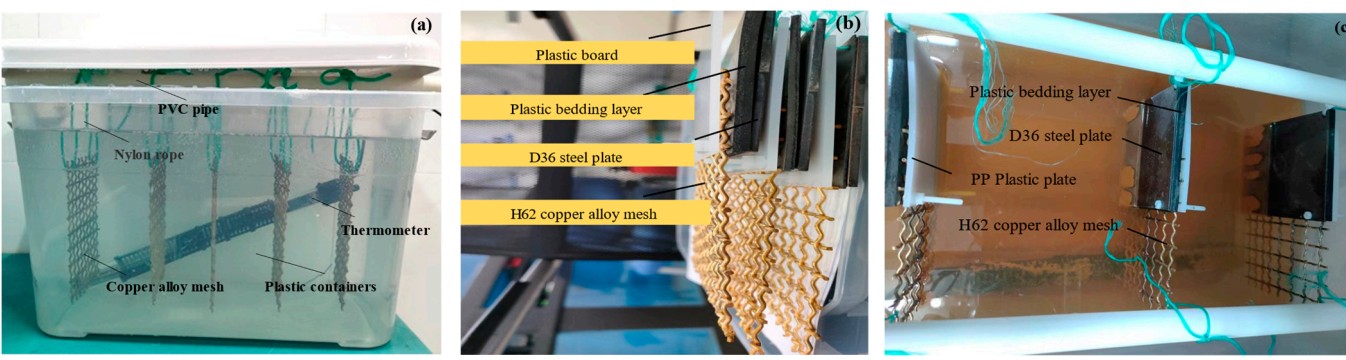

**Figure 4.** Diagrams of galvanic corrosion experiment. (**a**) Diagrams of full-immersed state. (**b**) Couple (with bedding layers). (**c**) Semi-immersed state for galvanic corrosion experiment.

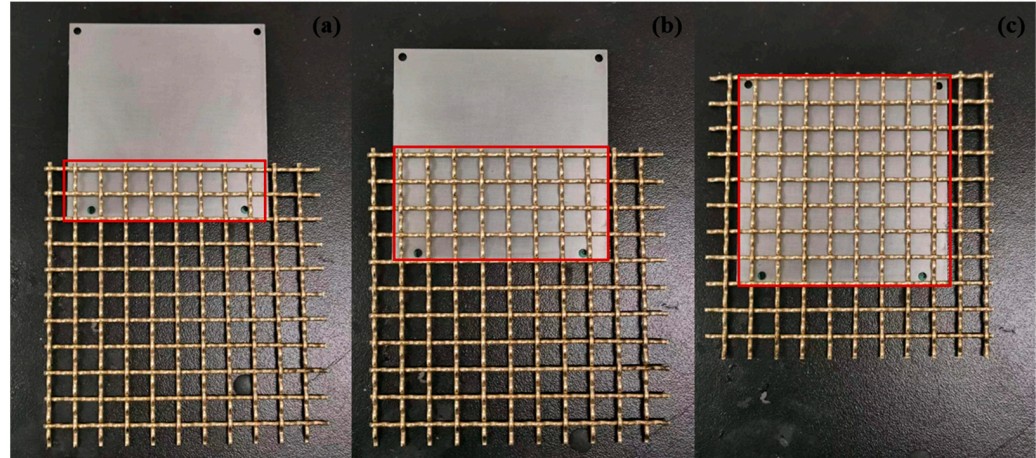

**Figure 5.** Diagrams of steel plate and copper alloy mesh overlapping. (**a**) 28 cm$^2$; (**b**) 52 cm$^2$. (**c**) 100 cm$^2$.

The galvanic corrosion rate was obtained by Formula (2):

$$K_c = \frac{(W_{c1} - W_{c0}) - (W_1 - W_0)}{St} \tag{2}$$

where $K_c$ is the average corrosion rate of the galvanic couple, mm/a; $W_{c1}$ is the mass of the anode group element coupled after the experiment; $W_{c0}$ is the mass of the anode group element coupled before the experiment, g; $W_1$ is the mass of the anode group element comparison specimen after the experiment, g; $W_0$ is the mass of the anode element comparison specimen before the experiment, g; $S$ is the experimental area of the anode element specimen, m$^2$; $t$ is the test time, h.

### 2.3.3. Electrochemical Test

CHI660 electrochemical workstation was used for the test system, which was a three-electrode system; the saturated calomel electrode was used as the reference electrode; the auxiliary electrode was a platinum electrode, and the D36 and H62 were the working electrodes. The electrolyte solutions were NaCl solutions with mass fractions of 1.5%, 2.5%, and 3.5%; the immersion states were full and half-immersion states, and the tests were conducted at room temperature. The galvanic couple polarization characteristic study was carried out in a 3.5% NaCl solution. The open-circuit point (OCP) test time was 1800 s.

When the OCP reached stability, electrochemical tests were conducted, and the linear polarization curve scan rate was 5 mV/s. The test frequency of EIS was $10^{-2}{\sim}10^{5}$, and the excitation signal amplitude was 10 mV. The electrochemical tests under every condition were conducted at least three times.

### 2.3.4. Tensile Test

A universal experimental tensile testing machine was used to test the mesh wire under different corrosion environments. In order to avoid the influence of external factors on the tensile test, the experiment was conducted at room temperature; the length of the mesh wire and the stretching speed were kept consistent. According to GB/T 228-2002, meshes would be corroded 240 h and 480 h. There are many factors that affect the results of tensile experiments, including the sample itself, the speed of stretching, the choice of the tensile test machine, the method of operation, and environmental factors. Therefore, a total of 10 wires dried for 24 h after different working conditions of the mesh were taken from different parts of the mesh in the uniaxial tensile test were.

Figure 6 is the fixture used in the mesh tensile test. The mesh wire diameter was the same, 100 mm. One end of the wire was placed in the lower fixed fixture groove; the other end was placed into the upper same fixture groove. In order to make the force uniform, the length of the wire placed in the two fixtures was approximately the same, and the force of 100 mm/min was applied to pull upward until the wire broke. Finally, the data was recorded and saved by the computer. The next set of experiments was conducted in the same way.

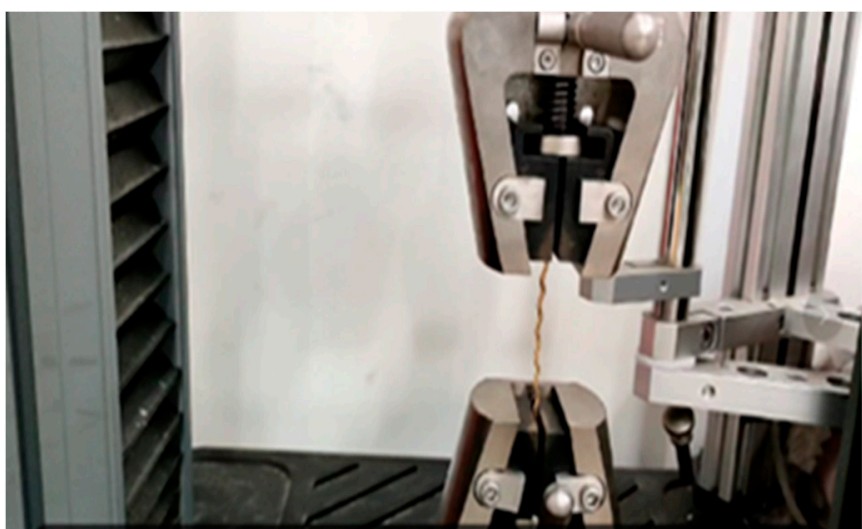

**Figure 6.** Clamp of tensile test.

### 3. Results

*3.1. Electrochemical Test*

3.1.1. Open Circuit Potential (OCP)

OCP is the trend of metal corrosion potential with time. The more positive OCP, the more difficult it is for the metal to lose electrons, so the more difficult it is to be corroded; the more negative open-circuit potential, the greater the thermodynamic corrosion tendency of the metal, and the easier it is to lose electrons. That is, it is easier to be corroded. The OCP of D36 steel and H62 copper alloy alone within 2000 s of immersion are shown in Figure 7.

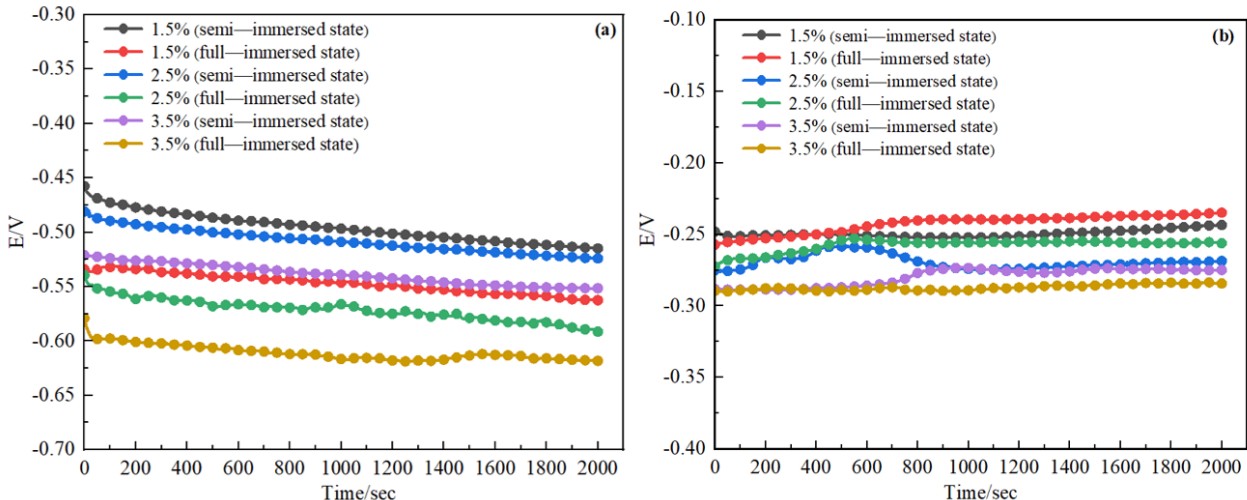

**Figure 7.** OCP of steel (**a**) and copper alloy (**b**).

The potential of H62 gradually moved in the positive direction and finally stabilized about −300 mV~−250 mV with time. D36 steel OCP gradually became negative and eventually stabilized about −600 mV~−450 mV. The difference between D36 steel and H62 copper alloy open circuit potential was about 250 mV after stabilization. According to the thermodynamic conditions of galvanic corrosion, the driving force for galvanic corrosion of D36 steel and H62 copper alloy was very large.

I-t curves of D36 steel and H62 copper alloy shown in Figure 8 showed the variation of current with time. It can be seen from the graphs that the current rose when the NaCl concentration increased, indicating that the NaCl concentration had a drastic effect on corrosion. The stronger the current, the faster the corrosion rate. However, the graphs did not clearly show that the semi-immersed state corrosion rate was greater than the full-immersed state or that the full-submersion state corrosion rate was greater than the semi-immersed state.

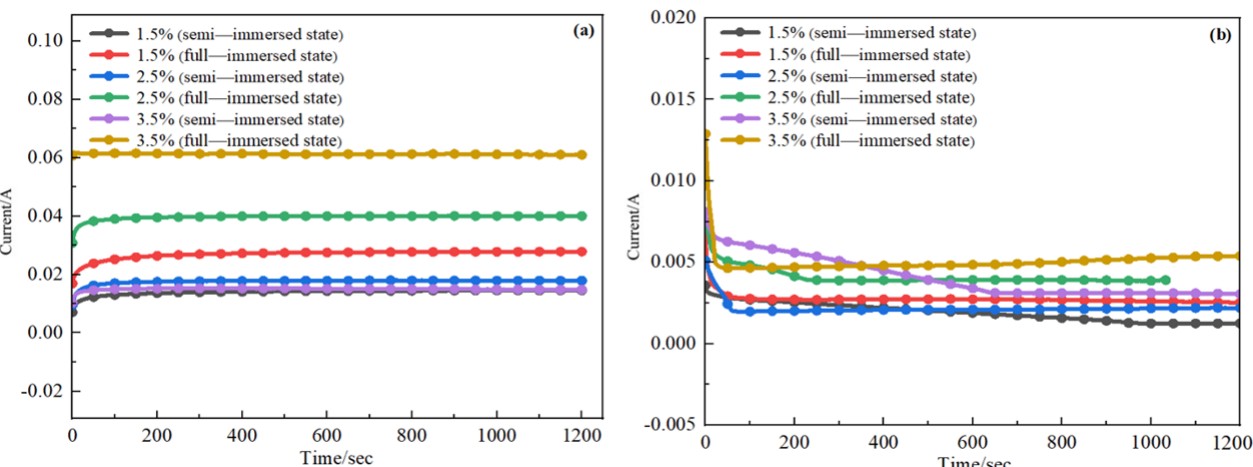

**Figure 8.** I-t curves of steel (**a**) and copper alloy (**b**).

### 3.1.2. Polarization Curve

In order to obtain the function relationship between the driving potential of the corrosion primary cell reaction and response rate current of D36 steel and H62 copper alloy under a simulated marine environment, the potentiodynamic polarization curves were tested.

Potentiodynamic polarization curves reflect the relationship between corrosion potential V and corrosion current density. The cathodic polarization curves and anodic



polarization curves are obtained by the Tafel extrapolation method. The potential and current corresponding to the intersection of cathodic and anodic Tafel line extension are the self-corrosion potential and self-corrosion current of the system. Self-corrosion potential reflects the ease of self-corrosion for the materials. The higher the self-corrosion potential, the higher the corrosion resistance of the material, and the more difficult it is to carry out corrosion. Self-corrosion current density reflects the material's open circuit potential corrosion rate. The greater the density, the greater the corrosion rate.

Table 3 shows the electrochemical parameters obtained by fitting in the electrochemical workstation CHI660E. As it can be seen from Figure 9, the anodic area of both curves does not appear to have obvious passivation. With the increase of NaCl concentration, both D36 and H62 were gradually decreasing for self-corrosion potential, and the self-corrosion current was becoming larger and larger. The self-corrosion potential of copper decreased from −291 mV to −473 mV, and the current density increased from −9.494 to −5.177. The self-corrosion potential of steel decreased from −570 mV to −673.5 mV, and the self-corrosion current density increased from −7.201 to −4.497. It indicated that with the increase of NaCl concentration, the corrosion rate of both materials was gradually increasing. In addition, copper alloy's self-corrosion potential was higher than D36 steel, and self-corrosion current density was lower than steel, indicating that under the same experimental conditions, the corrosion rate of the copper alloy was lower than steel, and corrosion resistance was larger.

**Table 3.** Electrochemical parameters of two materials.

| Material | Conditions | E/V | Current Density/log [I/(A·cm$^2$)] |
|---|---|---|---|
| H62 | 1.5% (semi-immersed state) | −0.291 | −9.104 |
| | 1.5% (full-immersed state) | −0.416 | −6.494 |
| | 2.5% (semi-immersed state) | −0.324 | −5.177 |
| | 2.5% (full-immersed state) | −0.437 | −5.793 |
| | 3.5% (semi-immersed state) | −0.460 | −5.386 |
| | 3.5% (full-immersed state) | −0.473 | −5.578 |
| D36 | 1.5% (semi-immersed state) | −0.570 | −7.201 |
| | 1.5% (full-immersed state) | −0.638 | −5.155 |
| | 2.5% (semi-immersed state) | −0.5907 | −5.171 |
| | 2.5% (full-immersed state) | −0.6655 | −5.683 |
| | 3.5% (semi-immersed state) | −0.596 | −4.708 |
| | 3.5% (full-immersed state) | −0.6735 | −4.497 |

**Figure 9.** Polarization curves of D36 steel (**a**) and H62 copper alloy (**b**).

### 3.1.3. Electrochemical impedance spectrum (EIS)

The Nyquist figures of copper alloy and steel were obtained by fitting the equivalent circuit with Zsimpwin software. As it can be seen from the Figure 10, H62 copper alloy and D36 all had only one capacitive resistance arc, indicating that there was only one time constant, the corrosive ions in the solution were subject to little diffusion resistance, and the electrode process was controlled by the charge transfer process. Moreover, the radius of capacitive arc resistance for copper alloy was much larger than that of steel, which indicated that in the simulated seawater solution, the impedance spectrum of the copper alloy was larger, and the corrosion resistance was better than that of D36 steel.

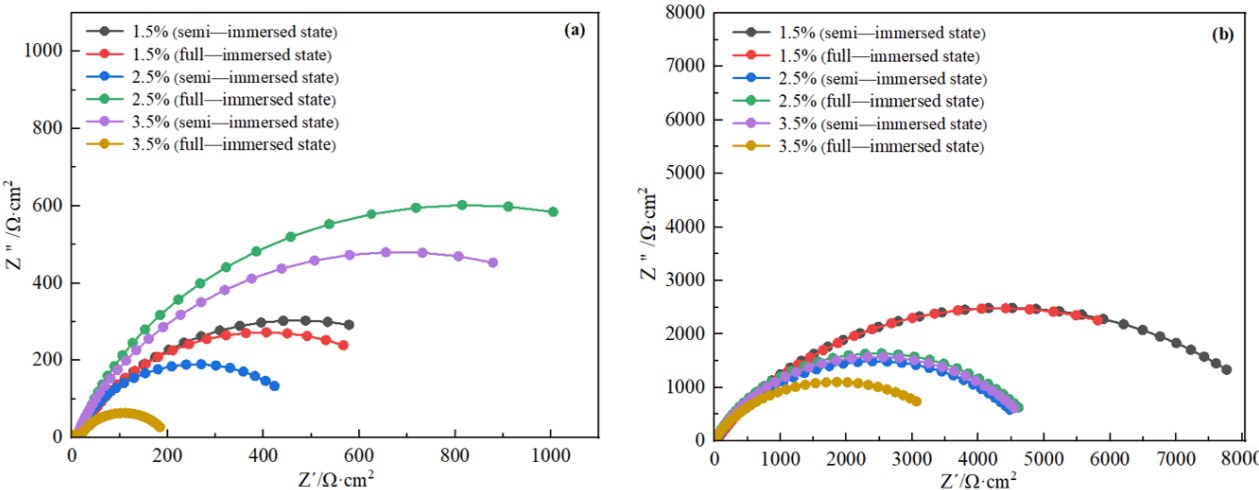

**Figure 10.** Nyquist plots of steel (**a**) and copper alloy (**b**).

The equivalent circuit shown in Figure 11 for the EIS test was used to fit the impedance spectrum, where $R_s$ is the solution resistance; $Q_1$ and $Q_2$ reflect the electric double-layer capacitance of corrosion product capacitance; $R_f$ is the corrosion product resistance; $R_{ct}$ is the charge transfer equivalent resistance of the double-layer; $n$ and $Y_0$ are the constant phase angle element CPE constant; $n$ is a reflection of the degree of dispersion, usually taking a value range of 0~1. The smaller the $n$, the more uneven the electrode surface microscopically and the greater the dispersion effect. When $n$ is 1, the capacitance is pure, and that is no dispersion effect. As can be seen from Table 4, both materials had oxide film resistance, indicating that both had the presence of oxide film on the surface, which had a certain barrier effect on corrosion. The charge transfer resistance $R_{ct}$ of the copper alloy was higher than steel, indicating the greater the resistance encountered in the corrosion process. Corrosion resistance was higher than steel, and copper alloy corrosion product film resistance was much higher than steel, indicating that the copper alloy surface-blocking corrosion ability was stronger; the material's corrosion-blocking ability was also better than steel.

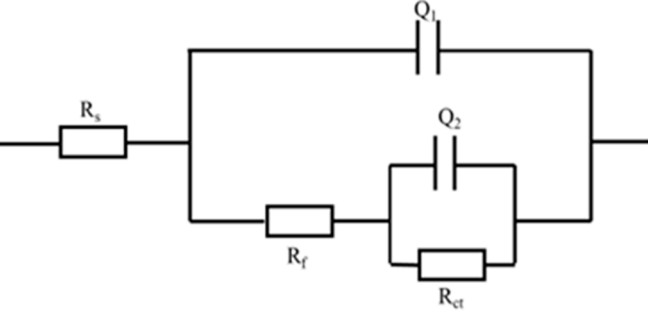

**Figure 11.** Equivalent circuit for EIS test.

**Table 4.** Fitting results of EIS tests.

| Material | Conditions | $R$ $(\Omega \times cm^2)$ | $Y_0$ $(S \times s^n/m^2)$ | $n$ | $R_f$ $(\Omega \cdot cm^2)$ | $Y_0$ $(S \times s^n/m^2)$ | $n$ | $R_{ct}$ $(\Omega \times cm^2)$ |
|---|---|---|---|---|---|---|---|---|
| H62 | 1.5% (semi-immersed) | 0.8332 | $2.88 \times 10^{-6}$ | 1 | 69.4 | $5.453 \times 10^{-5}$ | 0.618 | 8945 |
| | 1.5% (full-immersed) | 0.7168 | $1.09 \times 10^{-6}$ | 1 | 25.14 | $2.245 \times 10^{-4}$ | 0.664 | 8622 |
| | 2.5% (semi-immersed) | 1.182 | $1.068 \times 10^{-5}$ | 1 | 15.81 | $8.706 \times 10^{-5}$ | 0.69 | 4870 |
| | 2.5% (full-immersed) | 0.6016 | $4.936 \times 10^{-6}$ | 1 | 96.18 | $4.893 \times 10^{-5}$ | 0.698 | 4921 |
| | 3.5% (semi-immersed) | 0.6992 | $5.811 \times 10^{-6}$ | 1 | 137.1 | $5.776 \times 10^{-5}$ | 0.683 | 4830 |
| | 3.5% (full-immersed) | 0.4737 | $5.007 \times 10^{-6}$ | 1 | 27.1 | $8.822 \times 10^{-5}$ | 0.643 | 3752 |
| D36 | 1.5% (semi-immersed) | 1.195 | $7.488 \times 10^{-6}$ | 0.992 | 32.45 | $7.027 \times 10^{-4}$ | 0.735 | 744.1 |
| | 1.5% (full-immersed) | 15.59 | $1.429 \times 10^{-3}$ | 0.765 | 819.3 | $6.906 \times 10^{-4}$ | 0.84 | 0.01215 |
| | 2.5% (semi-immersed) | 0.4326 | $8.764 \times 10^{-6}$ | 0.965 | 9.394 | $1.726 \times 10^{-3}$ | 0.824 | 501.9 |
| | 2.5% (full-immersed) | 0.4952 | $5.479 \times 10^{-6}$ | 0.995 | 6.337 | $1.48 \times 10^{-3}$ | 0.794 | 757.4 |
| | 3.5% (semi-immersed) | 7.697 | $7.7 \times 10^{-4}$ | 0.771 | 38.64 | $6.872 \times 10^{-5}$ | 0.973 | 1313 |
| | 3.5% (full-immersed) | 0.9896 | $3.065 \times 10^{-6}$ | 0.973 | 16.01 | $2.274 \times 10^{-3}$ | 0.777 | 182.2 |

### 3.1.4. Galvanic Potential and Galvanic Current

Respectively, Figures 12 and 13 show the changes of galvanic potential and galvanic current with time for H62/D36 coupled with different layer thicknesses and different overlapping areas. From the figures, it could be seen that the galvanic potential increased with the increase of the bedding thickness and increased with the increase of the overlapping area. Meanwhile, the galvanic current was inversely proportional to the bedding thickness and the overlapping area. This indicated that adding the bedding layer or increasing the overlapping area between the copper mesh and the steel plate effectively reduced the corrosion rate, protecting the D36 plate.

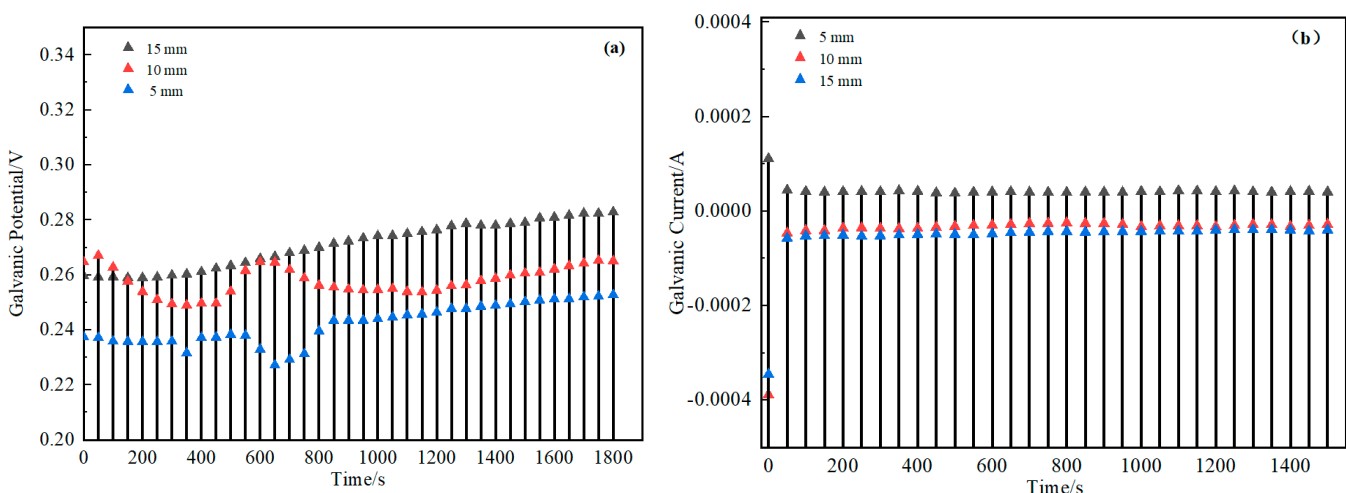

**Figure 12.** Galvanic potential-time (**a**) and galvanic current-time (**b**) plots of the two mate rials with bedding layers.

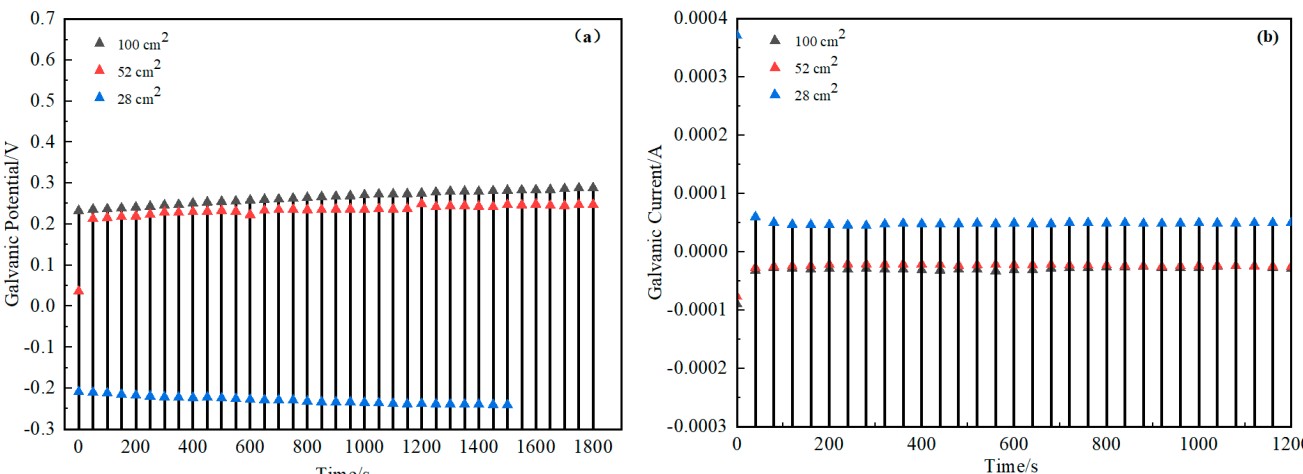

**Figure 13.** Galvanic potential-time (**a**) and galvanic current-time (**b**) plots of the two materials with overlapping areas.

*3.2. Corrosion Morphology and Mass Loss*

3.2.1. Corrosion Morphology

Figure 14 shows the macroscopic morphologies of copper alloy corroded at 120 h, 240 h, and 480 h in different concentrations of NaCl solution (1.5%, 2.5%, and 3.5%) under the semi/full-immersed state.

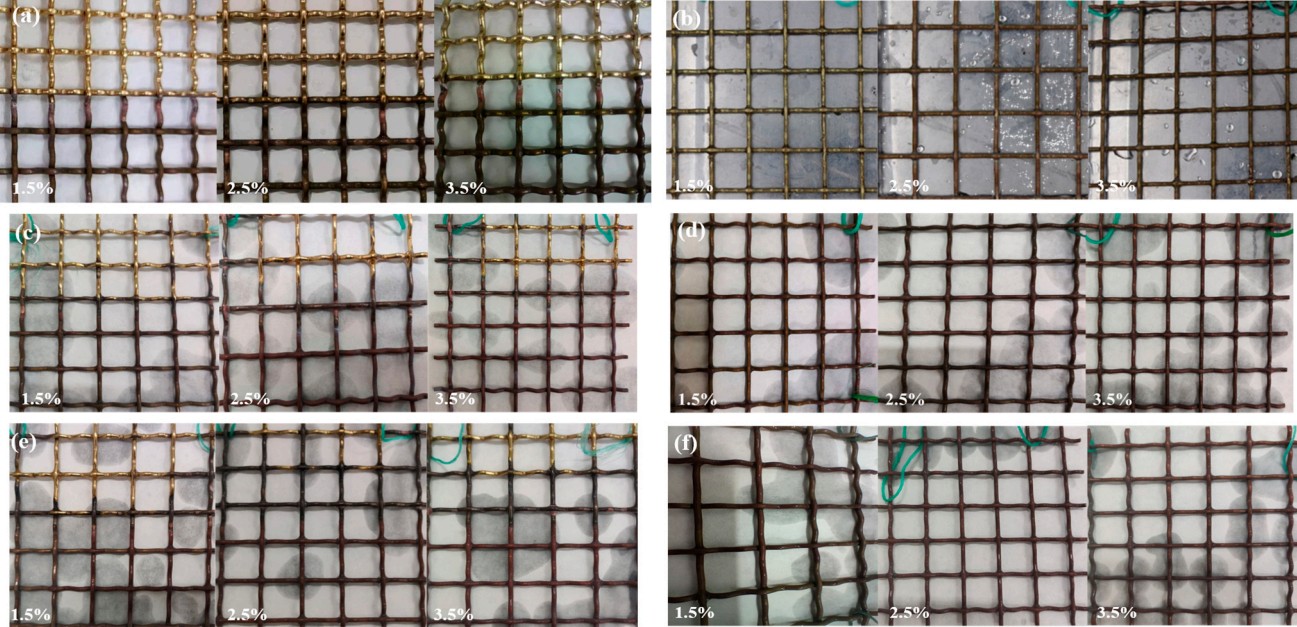

**Figure 14.** Macroscopic morphologies of the copper alloy for 120 h (**a**,**b**), 240 h (**c**,**d**), 480h (**e**,**f**) under the semi-immersed (**a**,**c**,**e**) and full-immersed (**b**,**d**,**f**) state.

As we can be seen from Figure 14, under the semi-immersed state, the H62 copper alloy mesh surface was smooth above the waterline part; the surface of the part below the waterline was reddish brown and red at the waterline in three different NaCl solutions after 120 h. After 240 h, the part at the waterline showed a clear reddish brown; the mesh below the waterline was brown, and this color was more obvious after 480 h. After 480 h, the part underwater was blackish. Under the full-immersed state, the original color of copper alloy mesh could still be seen after 120 h, although spots appeared on a few parts of the surface. However, with the increase of corrosion time, its surface color gradually changed to red–brown until black. Moreover, with the increase of NaCl concentrations,

surface black spots also gradually increased. Under the two immersion states, the H62 copper alloy surface lost gloss gradually and was not attached to the obvious corrosion products with time and the increase in concentrations. Corrosion products precipitated in the corrosion solution, mainly due to copper alloy corrosion film that plays a role in further corrosion. The existence of corrosion spots might be the local destruction of the surface layer with the protective film leading to the entry of corrosive media, weakening the protective effect formed.

It can be seen from Figure 15 that the macroscopic morphologies of D36 corroded at 120 h, 240 h, and 480 h in different concentrations of NaCl solutions (1.5%, 2.5%, and 3.5%) under semi-immersed/full-immersed state. It can be seen from the figures that there were significant changes at the waterline of D36 steel under the semi-immersed state, and the waterline was more and more obvious with time. The local area of the steel surface appeared to have raised corrosion product accumulated with time, scattered distribution, and difficult to remove with a brush. After the chemical treatment, the particles of the original surface on the steel plate changed into pitting pits of different sizes. D36 steel plates were uniform corrosion under the full-immersed state. In both submersion modes, there were two corrosion product layers. The first layer was the red corrosion product layer, and most of the corrosion product was loose and easy to fall off, precipitating in the corrosion medium. The second black corrosion products layer was removed with a soft brush; the surface with numerous small particles lost its original luster and became rougher with time.

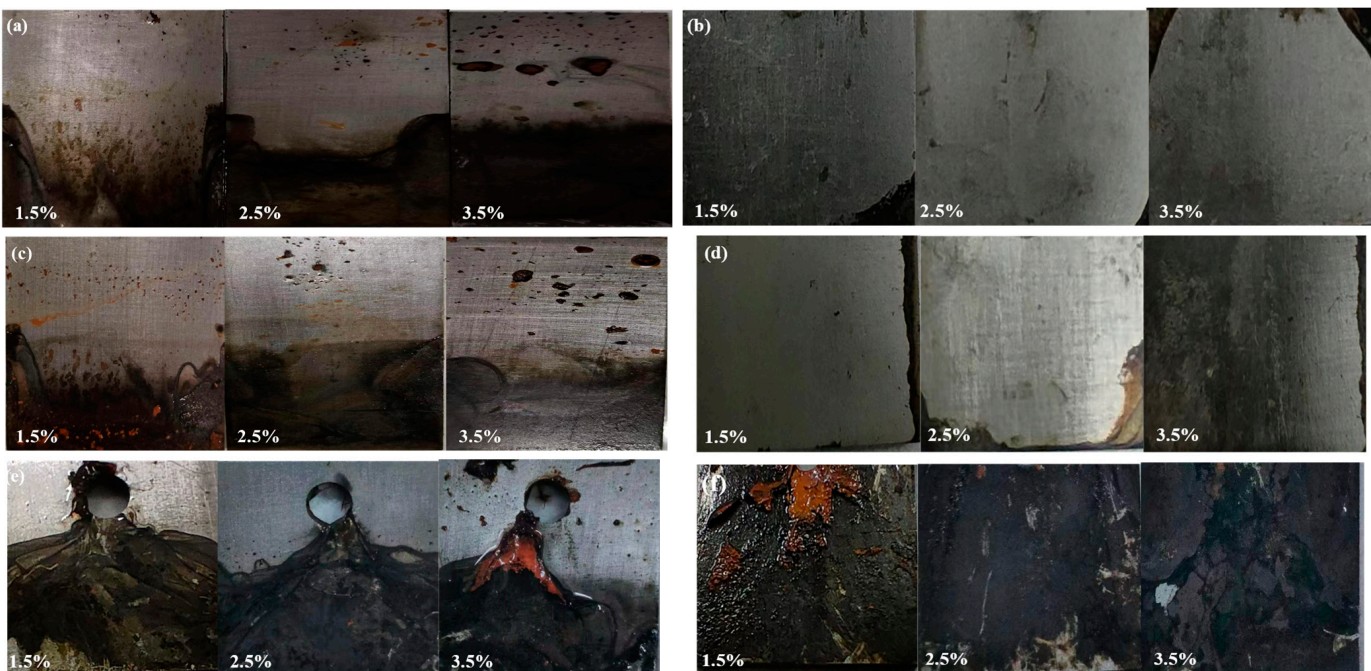

**Figure 15.** Macroscopic morphologies of the steel at 120 h (**a**,**b**), 240 h (**c**,**d**), 480 h (**e**,**f**) in semi-immersed (**a**,**c**,**e**) and full-immersed (**b**,**d**,**f**) state.

Figure 16 illustrates that with the increase of layer thickness and overlapping, the accumulation of corrosion products on the steel plate surface gradually decreased. When the layer thickness was 15 mm, there were no corrosion products near the waterline and above the steel plate, and there were almost no corrosion products at the other positions. The waterline was more and more obvious with time; the color was dark red, and gradually deepened.

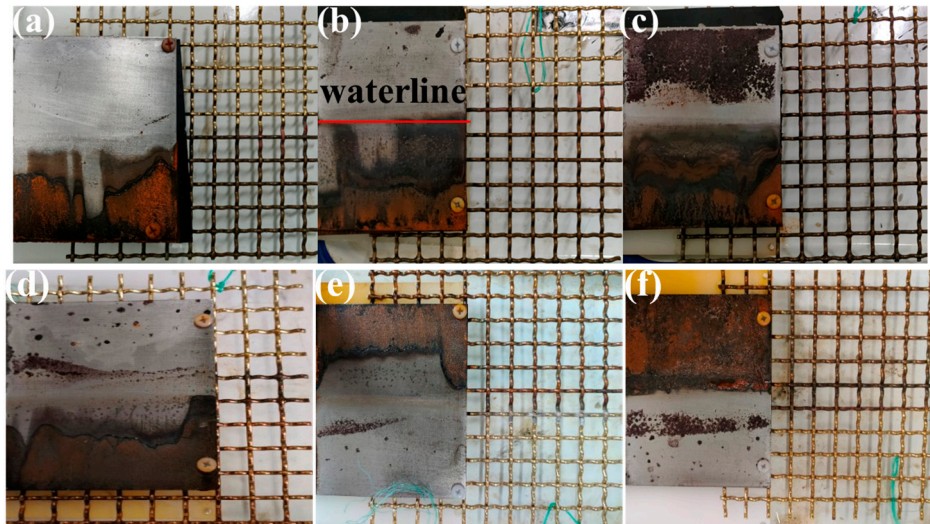

**Figure 16.** Graphs of galvanic corrosion with bedding layer (**a**–**c**) and overlapping area (**d**–**f**). (**a**) 5 mm. (**b**) 10 mm. (**c**) 15 mm. (**d**) 28 cm$^2$. (**e**) 52 cm$^2$. (**f**) 100 cm$^2$.

### 3.2.2. Weight Loss Analysis

Figures 17 and 18 show that the copper alloy corrosion rate increased faster in the initial stage but decreased at the later stage, mainly because the substrate was completely exposed without any protective measures at first, and the corrosion was relatively faster. However, a layer of oxide film formed on the metal surface with the prolongation of corrosion time. The protective film's charge transfer resistance improved, playing a hindering role in further corrosion. Seawater is an electrolyte with high salinity. The higher the salinity, the higher the electrical conductivity; the lower the oxygen, the less likely the possibility of forming a stable passivation film, and the higher the corrosion rate. Therefore, the corrosion rate of copper alloy and steel increased when the NaCl concentration increased. Yu et al. [25] also studied the effect of different temperatures and NaCl concentrations on H62 copper alloy and found that NaCl concentration was an important factor in the corrosion of H62 copper alloy, and its importance was higher than that of temperature. They also found that the steel corrosion rate was much higher than that of copper alloy in each condition, which was related to the different potential order of metal materials. D36 steel potential was more negative than copper; the stronger the activity, the greater the probability of losing electrons, and thus, the more likely steel was to be corroded, which was consistent with the electrochemical test results.

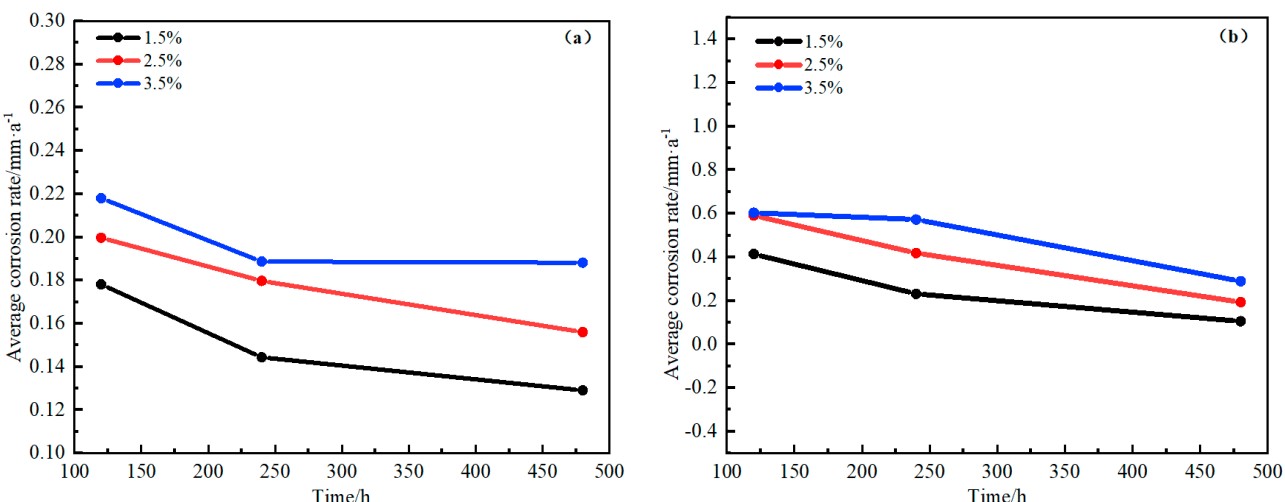

**Figure 17.** Changes of corrosion rate with time for D36 steel plates in different NaCl solutions. (**a**) Semi-immersed state. (**b**) Full-immersed state.

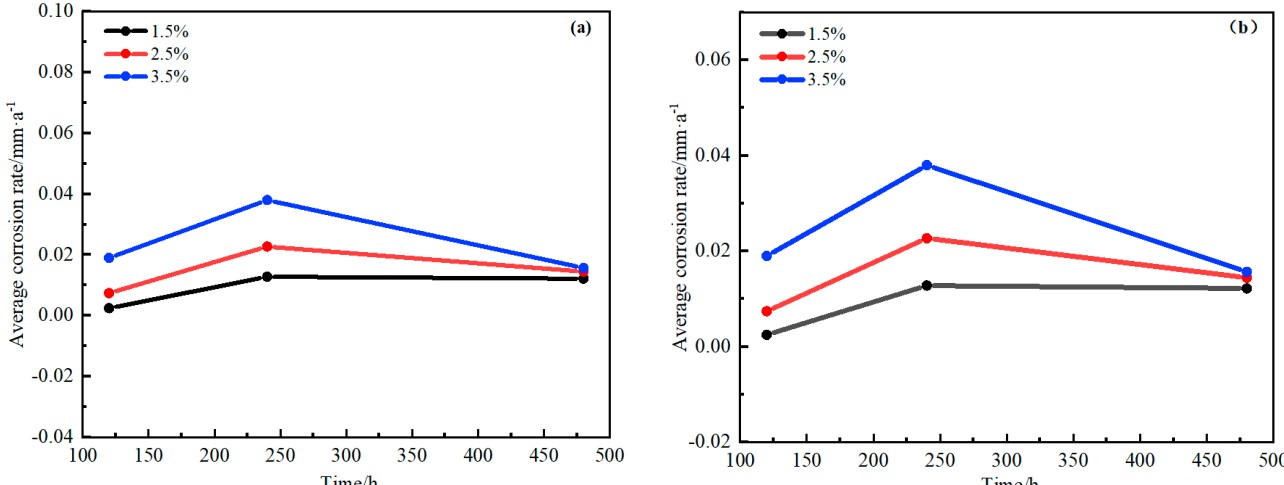

**Figure 18.** Changes of corrosion rate with time for H62 copper alloy mesh at different NaCl solutions. (**a**) Semi-immersed state. (**b**) Full-immersed state.

Figure 19 shows that the galvanic corrosion rate changed over time with layers or overlapping. It could be seen from the figures that the rate of galvanic corrosion increased with time. However, at the same time, the galvanic corrosion rate also increased slowly with the increase of the layer thickness. Similarly, the effect of the overlapping area on the galvanic corrosion rate was the same as that of when the layer thickness was added. This showed that the increase in the thickness of the layer and the increase in the overlapping area had played a certain protective role on the steel plate and delayed the galvanic corrosion rate.

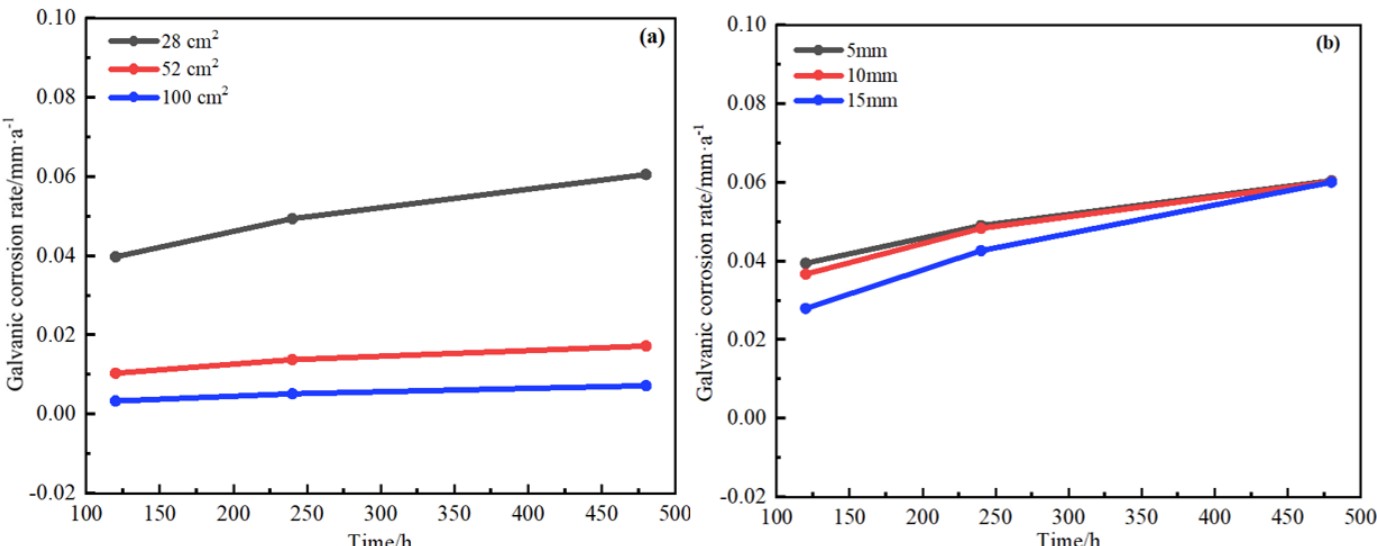

**Figure 19.** Changes of galvanic corrosion rate with time for H62 copper alloy meshes/D36 steel with overlapping area (**a**) and bedding layer (**b**).

### 3.3. Variation of Mechanical Properties of Copper Alloy Mesh

The net in the marine environment has to suffer not only from the damage of corrosion but also from the natural environment of strong winds and waves, which force the net to be subjected to tension in different directions. The tensile testing machine can apply external force to the sample to deform it and get relevant data. In this paper, the maximum force and tensile strength of each mesh wire were obtained, and the average value was taken to form the graphs about changes in the maximum force and tensile strength with corrosion time.

Figure 20 shows the broken area of mesh wire after corrosion under the semi-immersed mode. It was at the waterline. Because the middle of the mesh was corroded by both simulated seawater and atmosphere under the semi-immersed state, the degree of corrosion was greater than the upper and lower parts of the water line and more fragile compared to the upper and lower parts, which formed a fracture at the waterline. The maximum force and tensile strength of the mesh under two submersion modes decreased with time, and the higher the concentration of NaCl, the lower the maximum force and tensile strength. The reason was that the NaCl concentration is an important factor affecting the corrosion of the mesh, and the higher the NaCl concentration, the more serious the corrosion, and the more fragile the mesh was, the easier it was to break. At the same time, the maximum force and tensile strength of the net under all working conditions were much lower than that of the mesh before corrosion.

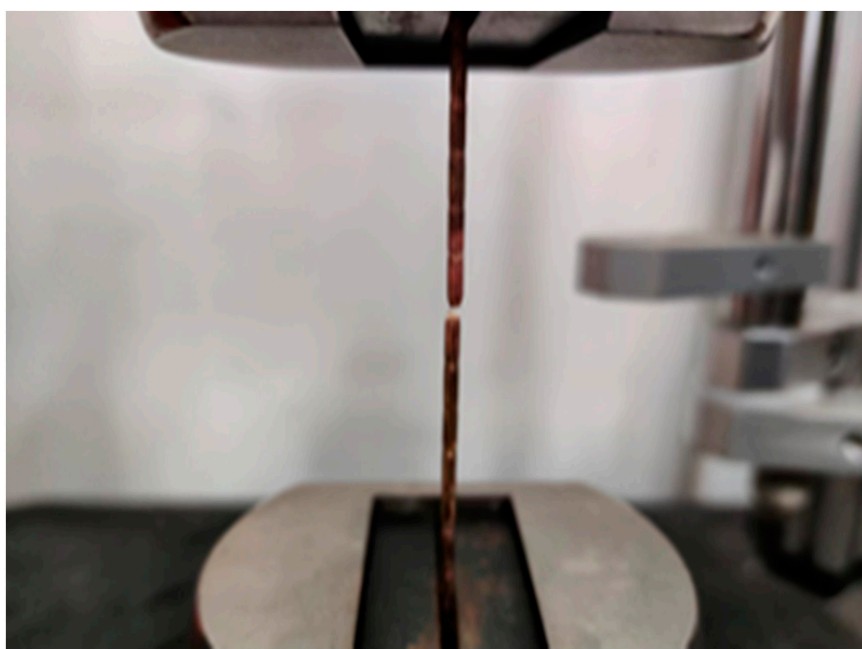

**Figure 20.** Broken area of mesh wire under the semi-immersed corrosion.

Figure 21a,b show the variations of the maximum force and tensile strength with time for the mesh when there are different bedding thicknesses between copper alloy and steel plate, where the maximum force was applied to the mesh when the bedding thickness was 0 mm. The reason was that the steel plate potential was lower and the copper alloy potential was higher, so the steel plate appeared as the anode in seawater when coupled with copper alloy mesh, which avoided further corrosion of the copper alloy to some extent, and the mesh could maintain a larger force. As the thickness of the bedding layer increased, the distance between the mesh and the steel plate increased, the anodic protection effect decreased, the corrosion rate of the steel plate decreased, and the corrosion rate of the mesh increased compared with the corrosion rate when it was directly lapped with the steel plate, so the force was gradually weakened when exposed in seawater. However, it could also be found that the bedding layer thickness was selected uniformly, the range was not large, and the results illustrated that force changes of the mesh due to the three bedding layer thicknesses were flat. The maximum force on the mesh gradually decreased with the extension of time, and all of them were lower than the maximum force of the mesh before corrosion. This showed that even though the mesh was directly lapped with the steel plate and the mesh was protected by the steel plate as the cathodic material, it still suffered some corrosion in the simulated seawater, which affected the force changes.

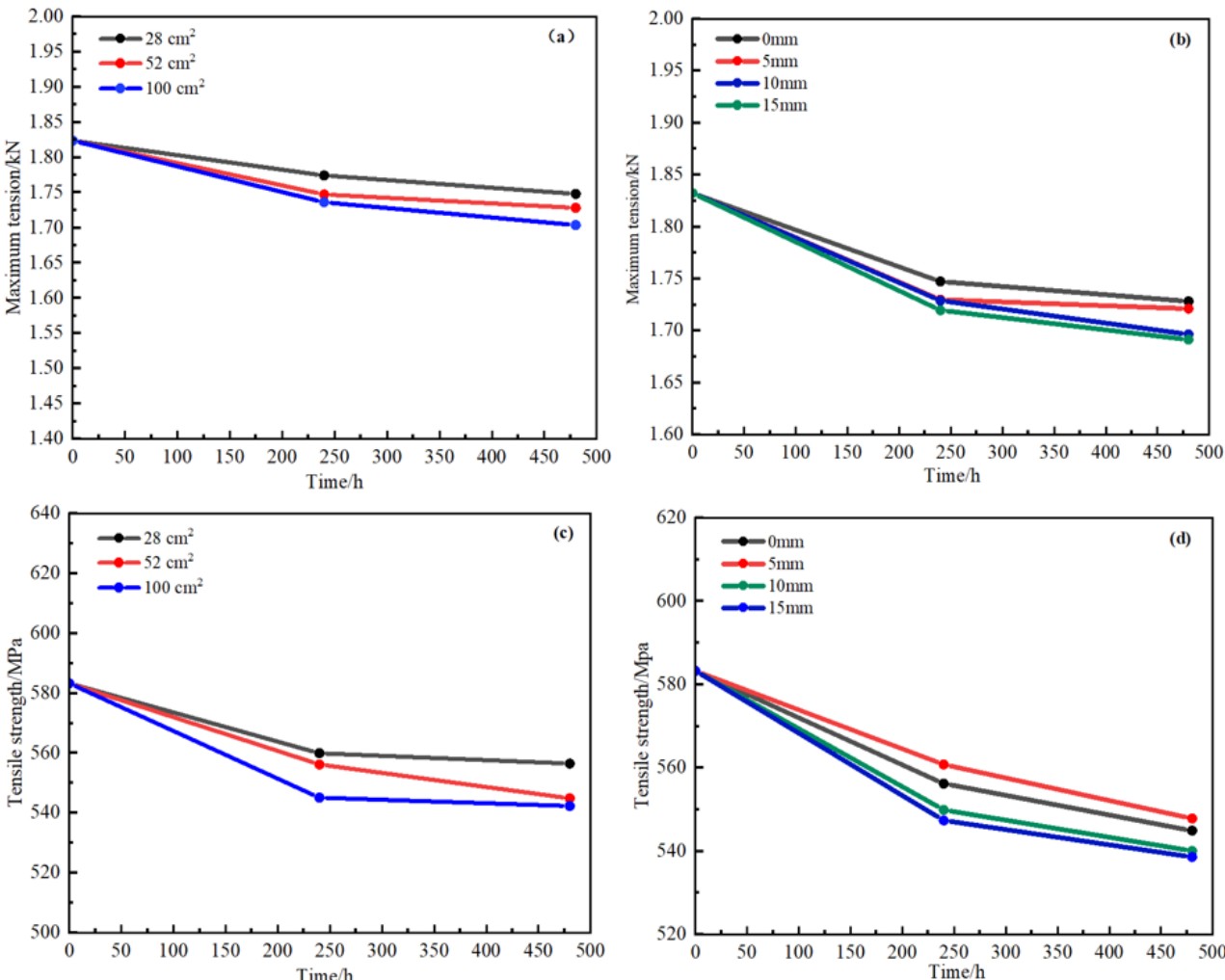

**Figure 21.** Variation of maximum tension (**a**,**b**) and tensile strength (**c**,**d**) for mesh with layer thickness (**b**,**d**) and overlapping area (**a**,**c**) with time.

Figure 21c,d show the variation of tensile strength of the mesh with bedding thickness and overlapping area. It was obvious that the tensile strength of the mesh was maximum at 0 mm, and the tensile strength gradually decreased with the increase of the layer thickness. The changes in the tensile strength for the mesh were gentle with time for three layers of thickness, which was maybe related to the selection of the bedding thickness. From the experimental results, it could be seen that the thickness of the bedding layer played a certain protective role in the marine aquaculture net cage frame steel. The greater the thickness of the bedding layer, the stronger the protection and the slower the corrosion rate.

## 4. Discussion

### 4.1. Effect of Salinity on Corrosion

Salinity is an important factor affecting corrosion. Generally, the salinity of the seawater surface is between 2.4% and 3.7%, and the salinity of deep seawater is between 3.4% and 3.5%; the difference is insignificant. The salinity in seawater directly affects the conductivity and oxygen content of seawater, which is positively proportional to the conductivity and inversely proportional to the oxygen content. NaCl, which accounts for 77.8% of seawater, is the main composition. Therefore, the effect of different NaCl concentrations on copper alloy mesh and a frame steel plate was investigated in this paper. The results of the study found that the corrosion rate of both copper alloy mesh and steel plate showed a trend of increasing and then decreasing in a short period of time. Meanwhile, the corrosion rate increased with the increase of salinity. Wang et al. [26] studied the corrosion behavior of

steel reinforcement in concrete by NaCl concentration electrochemical tests and revealed that pitting corrosion of reinforced concrete was positively correlated with the NaCl ion concentration in the environment. Li et al. [27] studied the effect of NaCl content on the pitting of cast and heat-treated diffusion-reinforced high-strength steels by electrochemical tests and analysis. The results showed that NaCl solution could reduce the dissolved oxygen content in solution with opposite results.

In addition, this paper studied the corrosion rate of mesh and frame steel plate in the simulated solution under a semi-immersed state and found that the changes were consistent with the results under the full submersion condition. Under the semi-immersed mode, both were subjected to the common corrosion of sodium NaCl solution and atmosphere at the same time, and the steel plate above the water line showed a different macroscopic appearance, with a large number of pitting pits of different sizes. Previous research on the impact of environmental factors on corrosion was focused on a full immersion test, concluding that the copper alloy and steel plate would show a uniform corrosion state. However, in recent years, marine engineering, especially mariculture platforms, in addition to the function of aquaculture, also increased recreation and other projects, so the semi-immersed type increased. The corrosion behavior of the mesh under a semi-immersed state provided data to supplement the corrosion data of steel plates and copper alloys and provided data support for the selection of materials under different types of aquaculture modes.

### 4.2. Effect of Layer Thickness on the Rate of Galvanic Corrosion of Copper Alloy and Steel

In order to improve the ability of marine aquaculture facilities to resist wind and waves, the frame and net are made of metal materials, and this material combination has been put into use in many countries. There's a vast sea area in China, and conditions of the sea are more complex and variable compared to other countries, so steel frame and copper alloy nets are used in facilities. However, the conditions under which galvanic corrosion occurs are mainly the potential difference of coupled metals, the presence of electron channels, and electrolytes. Copper alloy potential is more positive, as the ocean exists as the electrolyte, and steel potential is more negative, which exists as the anode, protecting the cathode in the seawater. The greater the potential difference between the two materials, the greater the driving force for galvanic corrosion. In this paper, we increased the thickness of the bedding layer to study the effect of the thickness of the bedding layer on the corrosion rate of the galvanic couple. The results showed that the greater the thickness of the bedding layer, the lower the corrosion rate of the steel plate, and the greater the corrosion rate of copper alloy. To a certain extent, the bedding layer thickness for the protection of the frame steel played a role. This research showed that the distance between the electric couple had an important influence on galvanic corrosion. When the cathode/anode area ratio is certain, the greater the distance, the greater the charged ion diffusion distance; the current density decreases, and the solution resistance increases at the same time.

Padding layers of different thicknesses were placed between steel plate and copper alloy mesh and then fixed with plastic PP plates and plastic screws to simulate the joint of the steel frame and copper alloy mesh, which widened the distance between the steel plate and copper alloy mesh to a certain extent and blocked the direct contact between them, but the range of the thickness of the mats obtained in this paper was small, so the changes of the results were flatter. In addition, in the seawater, the two materials overlapped under the semi-immersed state, and the simulation experiments in the laboratory were also mainly studied under the semi-immersed mode. The results presented in the semi-immersed state were also consistent with the full-immersed state. Sun et al. [28] studied the impact of cathode/anode area ratio and galvanic couple spacing on the galvanic corrosion of titanium alloy/steel in a 3% NaCl solution and found that when the electric couple spacing was reduced by 50%, the corrosion rate would instead increase by 34 to 45 times. From the above results, it can be seen that in the marine environment, whether it is a semi-immersed marine platform or a full-immersed facility, increasing the galvanic couple space with insulating

materials, blocking the direct contact of the electric couple, increasing the diffusion distance of the anion, and reducing the galvanic current density can largely slow down the galvanic corrosion rate and increase the service life of the anode materials.

### 4.3. Effect of Overlapping Area on the Rate of Galvanic Corrosion of Copper Alloy and Steel

There are many reports about the influence of the cathode/anode area ratio on the galvanic corrosion of galvanic couples. The cathode/anode areas ratio is considered to be an important factor affecting the corrosion of galvanic couples. Wu et al. [29] studied the galvanic corrosion behavior of 15 couples of ship metal materials with different cathode/anode area ratios and found that as the area ratio increased, the galvanic corrosion rate showed a trend of acceleration. They found that the larger the area ratio of the couples, the more serious the anode metal corrosion and the more obvious the phenomenon; the cathode material was protected from corrosion, and the corrosion rate was significantly lower than the rate of the presence alone. Jimmy et al. [30] studied the effect of geometry on the AZ91D/steel electric couple corrosion, also obtaining the conclusion that the larger the area, the greater the effect of electric couple corrosion. In the construction of the offshore platform, it always happens that choosing a material with a lower potential relative to the precious metal material as the anode to form a galvanic couple, making electrons continuously flow out from the anode into the cathode, and finally achieving the purpose of protecting the cathode. In this paper, according to the operation characteristics of the net cages under real marine conditions, the frame steel and copper alloy mesh were formed into a galvanic couple and directly increased or reduced their contact by continuously changing the overlapping area of the two materials in the simulated seawater under semi-immersed state to achieve the purpose of protecting the frame steel. The experimental results showed that with the increase of copper alloy/steel overlapping area, the corrosion rate of galvanic couples still increased with the increase of the overlapping area, but the corrosion rate was more moderate, indicating that the increase in "large anode, small cathode" area ratio played a certain protective role for the marine aquaculture frame steel.

Whether it is to increase the thickness of the layer or expand the overlapping area, the purpose is to reduce the rate of galvanic corrosion, reduce the impact of galvanic corrosion on the cathodes, extend the service life of the materials in the marine environment, promote the smooth operation of marine engineering, and achieve the efficient development of the marine economy.

### 4.4. Shortcomings and Further Prospects

The experiments in this paper are still inadequate, mainly because of the following: (1) The flow rate is an important factor affecting the seawater corrosion morphology and rate. In this paper, all the test conditions are in the static simulation of seawater, and China's coastal seawater flow rate can reach about 4 m/s; some materials also have a critical flow rate, so it is necessary to study the impact of different flow rates on the corrosion rate; (2) In the real marine environment, the marine aquaculture facilities are influenced by seawater corrosion and wave at the same time; the tension test in this paper was conducted after corrosion tests, then dried and placed over 24 h, and the results of the tests had a certain error.

Standard structures of seawater in the marine environment for the five zones: the marine atmosphere zone, seawater splash zone, seawater tidal difference zone, seawater full immersion zone, and submarine mud zone. The five zones have different degrees of corrosion impact, and salinity differences are also very large; the environment is more complex. Laboratory experiments are relative to real sea experiments, and the influence factor setting is single. The next research should focus on hanging tests under the semi-immersed state in the real culture area and comparing them with indoor tests. A single mesh test not only avoids the waste of materials caused by the whole net cage to do experiments but also obtains reliable data on the impact of corrosion.

The demand for protein and quality from the population and the support of national policies have driven the growth of the mariculture market, and the development of deep-sea aquaculture is unstoppable. The defects of existing net materials provide an opportunity for "copper" to develop. The studies on the corrosion behavior of single copper alloy mesh and copper alloy mesh/frame steel galvanic couple provide data support for the selection and protection of environmentally friendly and low-cost materials for aquaculture platforms, which is conducive to the green development of marine aquaculture.

## 5. Conclusions

In this paper, the corrosion rate of H62 copper alloy mesh and D36 steel plate of different salinity (full-immersed/semi-immersed state) and the effect of different bedding thickness and overlapping area on the galvanic corrosion rate of H62/D36 steel plate galvanic couples were studied, and tension experiments on the mesh under different working conditions to observe the changes of the maximum force and tensile strength for the mesh wires after being corroded were also conducted. The conclusions are summarized as follows:

(1) Salinity is an important factor affecting the corrosion of metals. In the simulated seawater solution, the corrosion rate of H62 copper alloy mesh and D36 steel plate also increased significantly with the increase of NaCl concentration. At the same time, the corrosion rate of both materials showed a trend of faster and then a slower trend in the short term with time. The OCP results showed that the copper alloy potential was more positive; the steel plate was negative, indicating that two materials had the tendency of galvanic corrosion (copper alloy was as the cathode, and the steel plate was as the anode material);

(2) The bedding layer thickness and the overlapping area both played a role in protection of the steel plate when the H62 mesh and D36 steel were coupled in seawater. With the increase of bedding layer thickness and overlapping area, the galvanic couple corrosion rate had a certain degree of reduction. Under the "large anode, small cathode" overlapping mode, the larger the overlapping area, the more moderate rise, although the corrosion rate was rising;

(3) The maximum force and tensile strength of the mesh in different corrosion conditions were also different. The smaller the concentration of NaCl solution, the smaller the corrosion rate of the mesh, and the greater the maximum force and tensile strength. At the same time, the maximum force and tensile strength of the meshes decreased when the thickness of the layer and overlapping area increased.

**Author Contributions:** Conceptualization, F.G. (Fukun Gui) and D.F.; methodology, F.G. (Fukun Gui) and F.G. (Fengfeng Gao); software, F.G. (Fengfeng Gao); validation, F.G. (Fengfeng Gao); formal analysis, F.G. (Fengfeng Gao); investigation, F.G. (Fengfeng Gao); resources, F.G. (Fukun Gui), X.Q. and F.H.; data curation, F.G. (Fukun Gui) and F.G. (Fengfeng Gao); writing—original draft preparation, F.G. (Fengfeng Gao); writing—review and editing, D.F and X.Y.; supervision, F.G. (Fukun Gui); project administration, F.G. (Fukun Gui); funding acquisition, F.G. (Fukun Gui) All authors have read and agreed to the published version of the manuscript.

**Funding:** This research was funded by the National Key Research and Development Project of China under grant number 2020YFE0200100, Key Research and Development Program of Zhejiang Province under grant number 2023C02029, the National Natural Science Foundation of China under grant number 32002441 and 42076213, Science and Technology Innovation 2025 Major Project of Ningbo City under grant number 2020Z076, and Zhoushan Science and Technology Projects under grant number 2022C01003. These financial supports are gratefully acknowledged.

**Institutional Review Board Statement:** Not applicable.

**Informed Consent Statement:** Not applicable.

**Data Availability Statement:** Data is available on request from the authors.

**Acknowledgments:** The authors would like to thank their advisors for their guidance and the reviewers for their constructive suggestions.

**Conflicts of Interest:** The authors declare no conflict of interest.

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
