# Peer review of "Study on the Corrosion Behavior of D36 Steel Plate and H62 Copper Alloy Net for Marine Aquaculture Facilities in Simulated Seawater"

_jmse, doi:10.3390/jmse11050975_

Round 1
Reviewer 1 Report
Comments and Suggestions for Authors
Paper title: Study on the corrosion behavior of D36 steel plate and H62 copper alloy netting for marine aquaculture facilities in simulated seawater
By Gao et al
General comments are given as follows:
The paper content is suitable for the specific topic of JMSE journal.
It is an interesting proposal for aquaculture and focus on net cages and how to reduce the corrosion rate. Many studies have been performed worldwide on the corrosion in offshore and marine structures.
Here it is a specific study on 2 metals (composition of metals also plays a role in corrosion). Results could interest the community but some improvements must be done to get a publishable and readable paper.
Comments are given versus the sections:
Title:
The title could be more “A laboratory study …..facilities” and suppressed “in simulated seawater”
Affiliations:
Line 7: where is the initials of F.F. Gao see page 24. This author has no contribution and he is the first one? Do not confuse FG and FG: FG1 and FG2.
Abstract:
Lines 14-16: not so clear.
Line 24: tensile results, please define clearly.
Keywords:
ok!
Introduction:
Line42: a space before “Facing…”.
Lines 56-57: it is evident!
Figure 1: some indications on photos would be appreciated by readers! What represents clearly a,b,c and d.
Line 75: “gotten” is “got”
Lines 75-79: authors must discuss about scale effect?
Lines 81,84, 109, 180 and after: all symbols must be written in all letters or explained before using the symbol.
Lines 89-90 :what is T2?
Line 98: put the symbol with brackets.
Line 111: it is figure 2! And avoid redundant term “supply”.
Figure 2: give on figures some scale or dimensions.
Table 1: last column: what means “Bal.”?
Table 2: For the first 2 lines: write parameter and under the unit
Line 122: is it direct tensile test?
Line 126: use number for subsection? As 2.2.1
Line 127: is it really a hole?
Lines 134, 146: “#”: what are the grain sizes?
Line 139: use number for subsection? As 2.2.2
Line 140: is it “seam” or “same”?
Line 142: use number for subsection? As 2.2.3
Line 152: use number for subsection? As 2.3.1
Line 158: what means “every other day”?
Line 160: cite the standard.
Figure 3: they are “photos” not diagrams. Give on photos some dimensions or indications.
Line 164: use number for subsection? As 2.3.2
Equation 1: it is ΔM not ΔW?
Line 168: write “where…” with small letter.
Lines 169 and 174: what is really “S”?
Figure 4: they are “photos” not diagrams. Give on photos some dimensions or indications. Not well placed here.
Figure 5: they are “photos” not diagrams. Give on photos some dimensions or indications.
Line 180: use number for subsection? As 2.3.3. what means “CHI660”? and line 248: CHI660E?
Line 186: what means EIS?
Line 191: use number for subsection? As 2.3.4. Direct tensile test.
Line 194: what means here “consistent”? Cite the standard in reference.
Line 195: 240h, 480h? what means this period?
Line 203: “approximatively”? Not too precise. Give one recorded curve and treatment for data.
Figure 6 not useful with any indication.
Figure 7: improve the readability. Write “s” as on other figures! On Y-axis: what is E: potential?
Figure 8: improve the readability. Write “s” as on other figures! On Y-axis: what is Current: intensity?
Line 224: What means I-t? write “gave the variation” not “showed”.
Line 227: a space before “The stronger…”.
Line 231: write “curve” in small letter.
Line 235: what means here “OPC”?
Figure 9: improve the readability. On Y-axis: what is E: potential?
Table 3: first line, correct “Material” and cm-2”.
.Line 247: check a space after “Table 3 ….”.
Line 248: see above remark.
Line 262: what means “D36 al?
Figure 10: improve the readability. Respect the same order of presentation: steel and copper as in other figures.
Line 269: suppress the brackets.
Table 4: in first 2 lines: write parameter in one line and under, the units. Use the same number of decimals for data (same accuracy). Improve the presentation in one page.
Figure 12: improve seriously the readability. Use the same color in a and b for symbols.
Figure 13: improve seriously the readability. Replace these figures 12 and 13 in the text.
Line 301: correct “under”.
Figure 14: improve seriously the readability of photos: these photos do not bring anything of interest because it is difficult to make difference in color. Reconsider the presentation. Add some indications.. Replace it in the text. Is it steel in legend?
Figure 15: improve seriously the readability of photos: these photos do not bring anything of interest because it is difficult to make difference in color. Reconsider the presentation. Add some indications.. Replace it in the text.
Figure 16: improve seriously the readability of photos: these photos do not bring anything of interest because it is difficult to make difference in color. Reconsider the presentation. Add some indications.. Replace it in the text.
Figure 17: improve seriously the readability of figures. Replace it in the text. Do not begin the subsection with figures. Write (a) and (b) before the legend or directly under the figure concerned.
Figure 18: improve seriously the readability of figures. Replace it in the text. Define a used in Y-axis. Write (a) and (b) before the legend or directly under the figure concerned.
Figure 19: improve seriously the readability of figures. Replace it in the text. Define a used in Y-axis. Write (a) and (b) before the legend or directly under the figure concerned. In legend, the order is (a) before (b).
Line 369: Put a “.” Before “But….”.
Line 377: “winds and waves….”.
Line 380: write “got”.
Figure 20: not useful, or combine with figure 6 (also not useful.
Figure 21: improve seriously the readability of figures. Replace it in the text. Correct in Y-axis: “kN” and “MPa”. The order used in legend is not clear for readers.
Lines 417-425: how authors do measurement of the section of wires for calculating tensile strength? Discussion could perform with non-dimensional data as ratio of strength compared to initial strength.
Line 428: “surface salinity” is salinity of water at surface.
Line 459: “to resist to winds and waves…”.
Line 493: “prolong” is not adequate term.
Line 521: put “,” after “cathodes”.
Line 524: “Shortcomings…”.
Lines 527-528: what is “flow rate…”?
Lines 529-533: Authors forget bacterial and microbiological effects.
Line 577: what is FF? where is FH? And FG is not distinguished?
References:
Check the writing in respect with guidelines for authors. It is not really an international review of references (more 50% are from the same country).
Reviewer 2 Report
Comments and Suggestions for Authors
The context from which the research presented in the manuscript is based is highly topical. Despite the high quality, I have several comments and recommendations for the authors:
Table 2: What was the basis for choosing the conditions of the experiments? Please justify.
Seawater concentration: What is the difference between real seawater and laboratory seawater? How do other ions affect corrosion activity?
Justify the meaning of ph, which you mention at the beginning of chapter 2.3 (line 155)
Figures 4 and 5 are not mentioned in the text. What is their informative value? Please complete the description in the text.
Chapter 3.3
What are the welding conditions in the production of the netting? How does the weld affect the corrosion-resistant properties of the tested materials in the netting form?
Reviewer 3 Report
Comments and Suggestions for Authors
This paper reports the corrosion rate of H62 copper alloy mesh and D36 steel plate of different salinity and the effect of different bedding thickness and overlapping area on the corrosion rate of H62/D36 steel plate galvanic couples.
The authors also conducted studies tension experiments on the mesh under different working conditions to observe the changes of the maximum force and tensile strength for the mesh wires after being corroded.
The paper "Study on the Corrosion Behavior of D36 Steel Plate and H62 Copper Alloy Netting for Marine Aquaculture Facilities in Simulated Seawater" is interesting and good structured. The methods and results presented in the manuscript are reliable.
In my opinion, in work missed:
1. The authors did not present the structures of the studied materials. The structure of the material has a very important impact on the course of corrosion.
2. Copper-zinc alloys in Cl-containing electrolytes are dezincification. This was also the case with the tested alloy H62. The authors do not state what the loss of zinc was, and how it affected the course of corrosion and tensile strength tests.
3. Fig. 14, 15, 16 are not very clear.
The paper should be checked against misprints and grammatical mistakes.
In summary, I think this paper can be published in Journal of Marine Science and Engineering after minor revisions.
Reviewer 4 Report
Comments and Suggestions for Authors
The article under review is a technical report on the practical problem of corrosion in marine environments. The authors conducted several tests, including OCP measurement, gravimetric tests, polarization tests, and material strength tests, to analyze the corrosion of materials used on nets in sea conditions. While most of the presented results are largely descriptive, the conclusions drawn confirm well-known observations regarding galvanic corrosion in the marine environment. Therefore, the biggest drawback of the article is the lack of any scientific novelties.
One area where the authors could improve the study is by addressing the issue of selective corrosion in the chosen alloy in marine conditions. Selective corrosion can have a significant impact on the durability and performance of materials in marine environments, and its effects should be thoroughly evaluated in any study of this nature.
Additionally, the article lacks information on the composition of the "bedding layers." If the layers are made of plastic, it is unclear whether there was electrical contact between the steel sheet and the copper alloy net.
Regarding the measurement techniques used in the study, the authors should provide a detailed explanation of how the current-time (I-t) curves were obtained, including how current was measured in the tested systems (i.e., steel and copper alloy). It would also be helpful to include a diagram of the tested system to aid in understanding the experimental setup and the flow of current through the system.
Furthermore, the authors should explain what galvanic potential is and how it is measured in Figures 12a and 13a. They should also provide an explanation of the measurement of the galvanic current shown in Figures 12b and 13b, including a diagram of the measurement system.
In conclusion, while the article presents a thorough analysis of the problem of corrosion in marine environments, the authors could enhance the study by addressing issues of selective corrosion, providing more detailed information on the experimental setup and measurement techniques used, and explaining key concepts and measurements in greater detail.
Author Response
请参阅附件

Round 2
Reviewer 4 Report
Comments and Suggestions for Authors
Thank you for the clear explanation and detailed photos of your experimental setups.
I am very sorry, but I am still confused.
According to the first photo in your answer (author_response.pdf), I cannot see the electric contact between the D36 steel plate and the H62 copper alloy mesh. So my question is whether the D36 steel plate and the H62 copper alloy mesh are connected through the electron conductor or if plastic elements isolate the two metallic parts very well.
During the experiments, when you measured the voltage in the cell and the current flowing, you shorted the two metal parts with an external conductor with crocodiles.
If so, you have lost all the protection given to you by the plastic parts, which are supposed to prevent galvanic corrosion. In my opinion, in this case, you artificially forced galvanic corrosion again.
Thus, it is difficult to say whether the data obtained from the above experiments better understand the effectiveness of bedding layers.
